# The Correlation of Arterial Stiffness Parameters with Aging and Comorbidity Burden

**DOI:** 10.3390/jcm11195761

**Published:** 2022-09-28

**Authors:** Francesco Fantin, Anna Giani, Monica Trentin, Andrea P. Rossi, Elena Zoico, Gloria Mazzali, Rocco Micciolo, Mauro Zamboni

**Affiliations:** 1Section of Geriatric Medicine, Department of Medicine, University of Verona, 37126 Verona, Italy; 2Centre for Medical Sciences, Department of Psychology and Cognitive Sciences, University of Trento, 38122 Trento, Italy; 3Section of Geriatric Medicine, Department of Surgery, Dentistry, Pediatric and Gynecology, University of Verona, 37126 Verona, Italy

**Keywords:** CAVI, CAVI0, PWV, comorbidity, aging

## Abstract

The aim of the study was to evaluate the relationships between carotid-femoral pulse wave velocity (PVW-cf), cardio-ankle vascular index (CAVI) and CAVI0 (which is a mathematical elaboration of CAVI, theoretically less dependent on blood pressure), age and comorbidity burden. Furthermore, 183 patients (119 female, mean age 67.5 ± 14.3 years) referred to the Geriatric Ward and Outpatient Clinic at Verona University Hospital were included; demographic, clinical and blood analysis data were collected. Charlson Comorbidity Index (CCI), PVW-cf, CAVI and CAVI 0 were obtained. Significant correlations were found between CAVI, CAVI0, PVW-cf and both age (r = 0.698, r = 0.717, r = 0.410, respectively *p* < 0.001 for all) and CCI, (r = 0.654; r = 0.658; r = 0.448 respectively and *p* < 0.001 for all), still significant after adjustment for several variables. In a stepwise multiple regression model, considering several variables, CCI was the only predictor of PWV-cf, whereas age and CCI were significant predictors of both CAVI and CAVI 0. In conclusion, all arterial stiffness indexes are associated with CCI and aging; the latter correlation is more evident for CAVI and CAVI 0 than for PVW-cf. Arterial stiffness parameters can complement the characterization of patients affected by a remarkable comorbidity burden across aging; arterial stiffening might mirror the complexity of these individuals.

## 1. Introduction

Vascular aging is associated with arterial wall remodeling, with progressive stiffening and reduced compliance; arterial stiffness is an independent predictor of cardiovascular morbidity and mortality [1]. It is therefore of remarkable importance to evaluate arterial stiffness in older individuals and in those adult patients who, owing to the presence of vascular and metabolic comorbidities, display high cardiovascular risk. Carotid-femoral pulse wave velocity (PVW-cf) and cardio-ankle vascular index (CAVI) are two common and feasible techniques aimed at detecting signs of vascular stiffening. As compared to PWV-cf, CAVI can evaluate arterial stiffness from a larger proportion of the arterial tree and is considered less dependent on blood pressure at the time of measurement [2,3]. Thus, although pulse wave analysis is considered the gold standard technique to evaluate vascular stiffness [4,5], CAVI can provide a more comprehensive assessment of arterial stiffness [3]. Furthermore, in order to further relieve the dependence of CAVI by blood pressure, in 2016 the mathematical expression of CAVI was elaborated and CAVI 0 was then suggested [6,7,8], and the association between CAVI and CAVI 0 has been widely demonstrated [9,10]. 

A massive number of pathological conditions have been shown to be related to increased arterial stiffening. For instance, PVW-cf is known to be associated to aging [11], cardiovascular risk factors [1] and metabolic syndrome [12]. On the other hand, increased CAVI is described in older subjects [13,14], in hypertensive patients [15,16], in the presence of vascular calcification and inflammation [17], in diabetic individuals and with concomitant metabolic diseases, [18,19], and in the presence of dyslipidemia [20,21]. Furthermore, weight loss is associated with CAVI reduction [22], and a positive association is described between CAVI and the presence of epicardial and visceral adipose tissue [23]. Increased CAVI is a predictor of cardiovascular events [24] and it is also associated to coronary artery disease [13], cerebral ischemia [25] and carotid arteries plaques [26].

Interestingly, although several comorbidities, as considered per se, are shown to be related to increased stiffness, less is known about the possible effect of the comprehensive comorbidity burden. The role of arterial stiffening in the characterization of complex patients with relevant comorbidity burden, considered across aging, is yet to be deeply explored; however, it may shed light on riveting pathophysiological issues. Furthermore, particular attention should be paid to cardiovascular comorbidities and risk factors, given their direct involvement in arterial structures. The aim of the study was to examine the correlation between arterial stiffness indexes, comorbidities and cardiovascular risk factors in a group of adults and older adults. 

## 2. Materials and Methods

The study population included 183 subjects, 119 females and 64 males, hospitalized at Geriatric Clinic of Verona University Hospital or referred to Outpatient Clinic (medical nutrition or arterial hypertension, of any age). Exclusion criteria were: (I) limb amputation or history of surgical treatment to aorta, carotids, or femoral arteries; (II) severe peripheral arterial disease or proximal arterial stenosis; (III) atrial fibrillation or other major arrhythmias. A detailed clinical history, with particular mention to cardiovascular diseases and risk factors, and physical examination were recorded for each patient. To evaluate the comorbidity burden, Charlson comorbidity Index (CCI) was calculated for each patient, using anamnestic patient-reported data. 

The study was approved by the Ethical Committee of the University of Verona. All participants gave informed consent to be involved in the research study.

### 2.1. Anthropometric Variables

With the subject barefoot and wearing light indoor clothing, body weight was measured to the nearest 0.1 kg (Salus scale, Milan, Italy), and height to the nearest 0.5 cm using a stadiometer (Salus stadiometer, Milan, Italy); whenever patients could not assume the erect position, the last anamnestic height was recoded. BMI was calculated as body weight adjusted by stature (kg/m^2^). 

### 2.2. Blood Pressure and Arterial Stiffness Measurements

CAVI, blood pressure and heart rate were measured and mean arterial pressure (MAP) and pulse pressure (PP) were calculated using VaSera-1000 (Fukuda-Denshi Company, LTD, Tokyo, Japan), as per the manufacturer’s recommendations. BP cuffs were placed simultaneously on the four limbs and inflated two by two (right and left side) to increase the accuracy of measurements. ECG was obtained by two electrodes placed on both arms; to obtain phonocardiography, a microphone was placed on the sternum (second rib space). This device calculates CAVI, on the basis of the Bramwell–Hill Formula [27,28], measuring heart-ankle PWV by the following equation:$$\mathrm{CAVI}=a*\left(ln\frac{P_{s}}{P_{d}}*\;\frac{{\mathrm{PWV}}^{2}*2\rho }{P_{s}-P_{d}}\right)+b$$
where *a* and *b* are constants, ρ is considered the blood density, *P_s_* stands for systolic blood pressure (SBP), and *P_d_* stands for diastolic blood pressure (DBP). By means of this device, heart-ankle PWV (haPWV) was calculated as the ratio between aortic valve to ankle length (automatically derived by software) and the time T taken by pulse wave to run this distance (T = tb + tba, tb = time from the second heart sound to the dicrotic notch at the brachial pulse wave form, tba = time from brachial to ankle pulse waves) [29]. CAVI 0 was derived by proper electronic calculator [30] following the formula:$$\mathrm{CAVI}\;0{\;}_{}=\frac{\mathrm{CAVI}-b}{a}*\;\frac{\frac{P_{s}}{P_{d}}-1}{\ln\left(\frac{P_{s}}{P_{d}}\right)}-ln\left(\frac{P_{s}}{P_{ref}}\right)$$
and considering Pref as a standard pressure of 100 mmHg.

### 2.3. Pulse Wave Velocity

The pulse wave analysis was performed noninvasively using a portable device called PulsePen (Diatecne, Milan, Italy) [31], and its software to obtain central aortic pressure values, an assessment of arterial pulse wave contours, an estimation of reflection waves and measurements of PWV. We previously provided a detailed description of PWA calculation, in previous studies [12,32]; we obtained carotid-femoral PWV (PWV-cf), which is considered representative of elastic arteries [33]. As recommended by consensus documents [34], the carotid-femoral distance was multiplied by a correction factor of 0.8. 

#### Biochemical Analysis

Venous blood samples were obtained after the subjects fasted overnight. Plasma glucose was measured with a glucose analyzer (Beckman Instruments Inc, Palo Alto, CA, USA). Cholesterol and triacylglycerol concentrations were determined with an automated enzymatic method (Autoanalyzer; Technicon, Tarrytown, NY, USA). High-density-lipoprotein (HDL) cholesterol was measured by using the method of Warnick and Albers. LDL cholesterol was calculated using the Friedwald formula [35]. Creatinine was measured by a modular analyzer (Roche Cobas 8000; Monza, Italy); eGFR was calculated by Cockroft–Gault formula.

### 2.4. Statistical Analyses

Results are shown as mean value ± standard deviation (SD). Variables not normally distributed were log-transformed before analysis. Pearson correlation coefficient was used to estimate correlations between variables. Independent samples t-tests were used to compare baseline characteristics of female and male patients. Analysis of variance (ANOVA) was performed when comparing continuous data, after stratifying the population upon age classes and comorbidities and to evaluate the effect of independent variables included in regression models. 

A significance threshold level of 0.05 was used throughout the study. All statistical analyses were performed using SPSS 23.0 version for Windows (IBM, Armonk, NY, USA). 

## 3. Results

The study population included 183 individuals, mean age 67.5 ± 14.3 years, 65% (*n* = 119) female. The main characteristics of the study population are listed in Table 1. 

### 3.1. Univariate Analysis

As shown by univariate analysis (Table 2), all arterial stiffness indexes display a positive relationship with age (CAVI r = 0.698, CAVI 0 r = 0.717, PVW-cf r 0.410, *p* < 0.001 for all of them). Furthermore, CAVI, CAVI 0 and PVW-cf resulted correlated to higher comorbidities, as measured by CCI (CAVI r = 0.654, *p* < 0.001; CAVI 0 r = 0.658, *p* < 0.001; PWV r = 0.448 and *p* < 0.001). Both CAVI and CAVI 0 showed a significant inverse relation with DBP (r = −0.296 and r = −0.389, respectively, *p* < 0.001 for both), MAP (r = −0.209, *p* = 0.005 and r = −0.274, *p* < 0.001, respectively) and a positive relation with PP (r = 0.165, *p* = 0.025 and r = 0.219, *p* = 0.003, respectively).

Moreover, CAVI 0 is directly correlated to CAVI (r = 0.955, *p* < 0.001) and both CAVI and CAVI 0 relate to PVW-cf (r = 0.430 and r = 0.438 respectively, *p* < 0.001 for both).

### 3.2. Subgroup Analysis: Cardiovascular Comorbidities and Risk Factors

As outlined by subgroup analyses, patients with hypertension diagnosis, as compared to patients without, had increased arterial stiffness indexes (mean PWV-cf 10.05 ± 4.67 vs. 8.63 ± 3.36, *p* = 0.017; mean CAVI 9.25 ± 2.13 vs. 8.19 ± 1.85, *p* = 0.001; mean CAVI 0 15.82 ± 6.57 vs. 12.95 ± 4.58, *p* = 0.003). Mean CAVI, CAVI 0 and PVW-cf were also increased in diabetic patients, when compared to normoglycemic subjects (mean PWV-cf 12.53 ± 5.42 vs. 9.013 ± 3.96, *p* < 0.001; mean CAVI 10.15 ± 2.50 vs. 8.64 ± 1.91, *p* < 0.001; mean CAVI 0 18.27 ± 7.31 vs. 14.11 ± 5.30, *p* < 0.001). Furthermore, the subgroup of patients with previous CV events, as compared to subjects without, had increased CAVI and CAVI 0, whilst PVW-cf was not significantly different between groups (mean CAVI 10.82 ± 2.46 vs. 8.65 ± 1.90, *p* < 0.001; mean CAVI 0 20.13 ± 7.52 vs. 14.18 ± 5.58, *p* < 0.001). When stratifying the study population upon CCI, as CCI increased, we outlined a progressive increase in CAVI (Figure 1A), CAVI 0 (Figure 1B), and PVW-cf (Figure 1C), which remained significant after adjustment for age, sex, MAP and GFR.

### 3.3. Regression Analysis: Arterial Stiffness Predictors

Stepwise multiple regression models were performed (Table 3) in order to evaluate the combined effect of independent variables on arterial stiffness parameters. In the first model PWV-cf was considered as a dependent variable; among several independent variables (age, GFR, MAP, CCI, LDL and triglycerides) only CCI resulted as significant predictor of PWV-cf (*p* < 0.001), accounting for 20.5% of its variance. Interestingly, when considering CAVI as dependent variable, and age, GFR, MAP, CCI, LDL and triglycerides as independent variables, both age and CCI resulted to be significant predictors (*p* < 0.001 and *p* = 0.012, respectively), explaining almost 53% of CAVI variance. Likewise, as shown in the third model which considered CAVI 0 as dependent variable and again age, GFR, MAP, CCI, LDL and triglycerides as independent variables, age and CCI (*p* < 0.001 and *p* = 0.010, respectively) could predict CAVI 0, accounting for 55.8% of its variance. 

## 4. Discussion

The present study shows significant positive correlations between all parameters of arterial stiffness and CV risk factors, comorbidities, and aging. The positive correlation with age is stronger for CAVI and CAVI 0 than PVW-cf. Moreover, our data confirm and complement previous knowledge showing that age and comorbidity can predict arterial stiffness parameters.

We could demonstrate a positive relationship between CAVI, CAVI 0, PVW-cf and the main CV risk factors, even after adjustment for age, sex, MAP and GFR. In line with previous evidence [36], in our population all the arterial stiffness indexes resulted increased in hypertensive patients, reflecting the vascular remodeling, characterized by wall stiffening typical of this condition. In particular, we outlined a significant increase in CAVI among hypertensive subjects, which is consistent with the results of Nagayama and colleagues [16], who demonstrated increased CAVI values in a cohort of 2300 individuals, describing a sharper increase after the SBP threshold of 140 mmHg.

As is predictable considering vascular involvement in the diabetes mellitus course [37], we described increased arterial stiffness indexes among diabetic patients and among subjects with impaired fasting glucose levels, as compared to normoglycemic individuals. These results are in line with previous finding regarding both PVW-cf [37] and CAVI [18,38]; the latter was found increased in diabetic patients, showing however a progressive decrease after 8 weeks of glucose lowering therapy, consistent with HbA1c reduction [18]. Moreover, we found a significant correlation between all the arterial stiffness indexes and metabolic syndrome components, confirming previous evidence [12,39,40], and corroborating the hypothesis of increased arterial stiffening as a crucial change in the presence of metabolic disorders or metabolic syndrome. 

We further outlined increased CAVI and CAVI 0 in patients with previous CV events, still in line with several studies that described increased CAVI in subjects with known coronary artery disease and cerebral ischemia [25,41,42,43]. Altogether, our and other results suggest that different vascular diseases, affecting different segments of the arterial tree, share the common finding of increased arterial stiffness. Our findings actually complement previous observations because they show that the heterogeneity of the vessels involved may be more accurate by testing CAVI, instead of PVW-cf, since the first is more representative of a large proportion of the arterial tree [3]. 

Although several conditions are known to be associated to worse values of CAVI and PVW-cf, the possible association with the comprehensive comorbidities burden is not entirely explored. In this regard, although less is known about increased comorbidity index, CAVI has already been depicted as increased among frail individuals; thus, our results confirm and complement previous evidence by Xue and colleagues who described higher CAVI values in elderly frail patients (relaying on Fried’s frailty definition) [44]. Noteworthily, moving one step further, we observed a positive relationship between all the arterial stiffness indexes and both comorbidities number and CCI, still significant after adjustment for age, sex, MAP and GFR. The pathophysiological background of this finding might rely on the vascular remodeling, which occurs during healthy aging [45], and pathological conditions [1]. Certainly, arterial stiffening is the common denominator of several diseases included in the CCI calculation, and therefore a double-sided relation might be inferred: first, arterial wall stiffening in otherwise healthy aging subjects might increase the risk of developing a huge number of vascular-associated conditions. On the other hand, presenting relevant comorbidities (primarily involving or not the vascular system) might promote a complex network of tissue remodeling processes, leading to arterial wall stiffening. Thus, more than a single disease, the comprehensive burden of multiple co-existing conditions might contribute to widespread and increased vascular stiffening. According to the latter interpretation, we could demonstrate that considering arterial stiffness parameters as dependent variables, the comorbidity burden described by CCI is a strong predictor of all the arterial stiffness indexes, and along with age it can explain a consistent percentage of CAVI and CAVI 0 variance. Arterial stiffness might thus be considered as part of the expression of a multidimensional decline; in particular we found that CAVI, as compared to PVW-cf, was more strongly related to the comorbidity burden, and, once more, a possible explanation lies in the wider proportion of arterial segments that is simultaneously investigated by CAVI, therefore including a broad spectrum of pathological conditions. 

According to consolidated knowledge [45], our study confirms a significant association between aging and arterial stiffness: each arterial stiffness index displays significant relation with age; the strength of the association is higher for CAVI and CAVI 0, as compared to PVW-cf. Furthermore, on top of several variables, age is shown to be a significant predictor of CAVI and CAVI 0, yet not of PWV-cf. The remarkable relationship between age and CAVI was previously demonstrated by Shirai et al. [46], who described increased CAVI in elderly subjects as a possible expression of age-related arterial wall sclerosis. Although PVW-cf is still considered as the primary arterial stiffness evaluation technique for outcome prediction [47], CAVI could be also considered as a more reliable index in the elderly, due to its lower dependence from blood pressure [48].

A few limitations of the study should be recognized: this is a cross-sectional study, and therefore we were not allowed to test the power of arterial stiffness indexes in predicating long term cardiovascular risk. Our study was predominantly performed in female patients and given the increased prevalence of cardiovascular diseases in the male population, we need to test our hypothesis in a more represented male population. Further, information regarding medical therapy was not available, but we acknowledge that the possible effect of different medications on arterial stiffness may be of interest. As concerns arterial stiffness parameters, we acknowledge that the augmentation index, which is deemed to be an important parameter, was not significant in our findings, and therefore excluded from the results.

In conclusion, our study, conducted on a relatively wide and heterogeneous cohort of patients, demonstrated that PVW-cf, CAVI and CAVI 0 are associated to CV risk factors and higher comorbidity burden, even after adjustment for several variables. Furthermore, our data outline a strong correlation between arterial stiffness indexes and age. Our findings might complement the pathophysiological understanding of the cardiovascular impairment in subjects with older age and remarkable comorbidity burden. Therefore, in the clinical setting, arterial stiffness evaluation, which is a feasible and easily available technique, may complement the characterization of complex patients. 

## Figures and Tables

**Figure 1 jcm-11-05761-f001:**
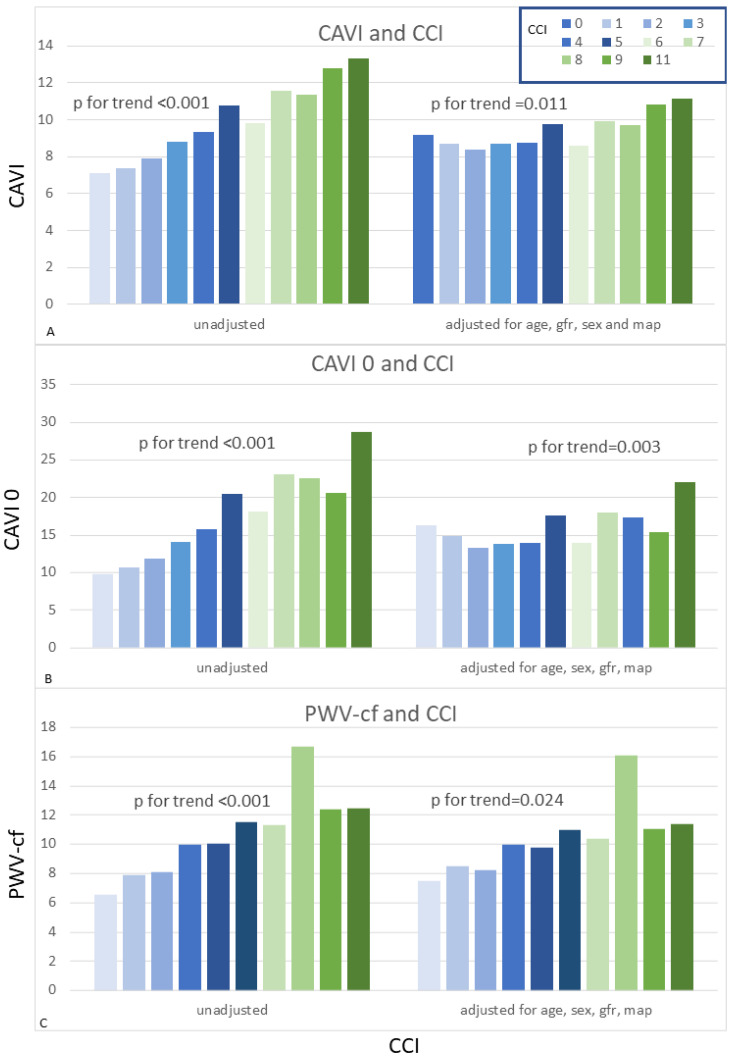
CAVI (**A**), CAVI 0 (**B**) and PWV-cf (**C**) values stratified by Charlson Comorbidity Index Columns represent increasing values of cci; on y axes arterial stiffness parameters are displayed.

**Table 1 jcm-11-05761-t001:** Main characteristics of the study population.

	Total (*n* = 183)	Male (*n* = 64)	Female (*n* = 119)	*p* Value
Age (years)	67.54 ± 14.25	70.13 ± 14.43	66.14 ± 14.02	0.075
Body weight (kg)	77.06 ± 18.01	81.30 ± 18.90	74.78 ± 17.17	0.023
BMI (kg/m^2^)	28.92 ± 5.85	27.77 ± 5.65	29.54 ± 5.88	0.048
Glucose level (mg/dL)	100.08 ± 26.26	104.74 ± 30.95	97.41 ± 22.89	0.104
Total Cholesterol (mg/dL)	179.43 ± 46.92	164.16 ± 46.19	187.74 ± 45.39	0.001
HDL Cholesterol (mg/dL)	50.81 ± 17.02	46.75 ± 16.18	52.95 ± 17.14	0.02
LDL Cholesterol (mg/dL)	105.23 ± 40.80	93.38 ± 40.26	111.43 ± 39.88	0.007
Triglycerides (mg/dL)	132.35 ± 67.13	132.84 ± 71.81	132.07 ± 64.68	0.944
Creatinine (mg/dL)	0.95 ± 0.44	1.09 ± 0.52	0.87 ± 0.36	0.001
GFR (mL/min/1.73 m^2^)	83.59 ± 36.90	85.15 ± 44.14	82.74 ± 32.52	0.675
SBP (mmHg)	138.84 ± 17.09	134.83 ± 16.40	140.99 ± 17.13	0.018
DBP (mmHg)	81.22 ± 10.85	80.81 ± 12.83	81.44 ± 9.67	0.711
PP (mmHg)	57.72 ± 13.940	53.86 ± 11.48	59.79 ± 14.73	0.003
MAP (mmHg)	110.03 ± 12.58	107.82 ± 13.54	111.21 ± 11.93	0.095
CAVI	8.92 ± 2.09	9.58 ± 2.23	8.56 ± 1.94	0.003
CAVI 0	14.93 ±6.16	16.20 ± 6.48	14.24 ± 5.89	0.047
PWV-cf (m/s)	9.58 ± 4.36	9.39 ± 3.45	9.69 ± 4.79	0.636
Number of diseases	5.42 ± 2.41	5.45 ± 2.34	5.39 ± 2.45	0.875
CCI	3.30 ± 2.24	3.72 ± 2.31	3.07 ± 2.18	0.066

BMI: body mass index, HDL: high density lipoprotein; LDL: low density lipoprotein; GFR: glomerular filtration rate; SBP: systolic blood pressure; DBP: diastolic blood pressure; PP: pulse pressure; MAP: mean arterial pressure; CAVI: cardio-ankle vascular index; PWV-cf: pulse wave velocity carotid-femoral; CCI: Charlson Comorbidity Index.

**Table 2 jcm-11-05761-t002:** Univariate Correlations between CAVI, CAVI0, PWV-cf and the main clinical variables.

	CAVI	CAVI 0	PWV-cf
Age	0.698 ***	0.717 ***	0.410 **
Glucose level	0.166 *	0.166 *	0.152
Total Cholesterol	−0.446 ***	−0.430 ***	−0.203 **
HDL Cholesterol	−0.187 *	−0.213 **	−0.173 *
LDL Cholesterol	−0.474 ***	−0.479 ***	−0.237 **
Triglycerides	0.036	0.070	0.139
GFR	−0.535 ***	−0.521 ***	−0.213 **
CCI	0.654 ***	0.658 ***	0.448 ***
SBP	−0.060	−0.074	−0.017
DBP	−0.296 ***	−0.389 ***	−0.146 **
MAP	−0.209 **	−0.274 ***	−0.097
PP	0.165 *	0.219 **	0.108
CAVI	1	0.955 ***	0.430 ***
CAVI 0	0.955 ***	1	0.438 ***
PWV-cf	0.430 ***	0.438 ***	1

* *p* < 0.05, ** *p* < 0.01, *** *p* < 0.001. HDL: high density lipoprotein; LDL: low density lipoprotein; GFR: glomerular filtration rate; CCI: Charlson Comorbidity Index; SBP: systolic blood pressure; DBP: diastolic blood pressure; MAP: mean arterial pressure; PP: pulse pressure; CAVI: cardio-ankle vascular index; PWV-cf: pulse wave velocity carotid-femoral.

**Table 3 jcm-11-05761-t003:** Stepwise regression analysis, considering PWV-cf, CAVI, and CAVI 0 respectively as dependent variables, and age, glomerular filtration rate, mean arterial pressure, Charlson Comorbidity Index, LDL-Cholesterol and Triglycerides as independent variables.

Dependent Variables	Independent Variables	β ± Standard Error	*p* Value	R^2^
PWV-cf				
	CCI	0.924 ± 0.144	<0.001	0.205
CAVI				
	Age	0.076 ± 0.014	<0.001	
	CCI	0.238 ± 0.094	0.012	0.528
CAVI 0				
	Age	0.226 ± 0.039	<0.001	
	CCI	0.680 ± 0.259	0.010	0.558

CCI: Charlson Comorbidity Index; CAVI: cardio-ankle vascular index; PWV-cf: pulse wave velocity carotid-femoral.

## Data Availability

The data presented in this study are available on request from the corresponding author.

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
