# Peer review of "The Correlation of Arterial Stiffness Parameters with Aging and Comorbidity Burden"

_jcm, 2022, doi:10.3390/jcm11195761_

Round 1
Reviewer 1 Report
With interest I have read the MS Arterial stiffness, aging and comorbidity by Fantin et al.
The study investigates the association between arterial stiffness indexes, comorbidities and cardiovascular risk factors in a group of adults and older adults.
The topic is interesting, the research group is known on this field and the study seems the well performed.
I have only some minor comments.
Please use the term ageing or aging consequently
Line 42 ‘CAVI can provide a more comprehensive assessment of arterial stiffness’ Please use a reference supporting this statement
Line 50 plaques (typo)
Please define how GFR or eGFR was defined
The study would improve if the authors would stress the clinical relevance of the findings. Probably one sentence on this would be sufficient.
Author Response
REVIEWER 1
Please use the term ageing or aging consequently
We agree with the reviewer, the term aging was chosen and consistently used throughout the text.
Line 42 ‘CAVI can provide a more comprehensive assessment of arterial stiffness’. Please add a reference supporting this statement.
As recommended, a reference was added. (now line 53, Ref 3).
Line 50 plaques (typo)
Corrected (now line 64).
Please define how GFR or eGFR was defined
This issue is now included in the method section; Cockroft-Gault formula was used. (line 118)
The study would improve if the authors would stress the clinical relevance of the findings. Probably one sentence on this would be sufficient.
We appreciate this recommendation, and as suggested a sentence (in Conclusion Section) was added to highlight the clinical implication of our study. (Line 233-236)

Reviewer 2 Report
Reviewer Report is attached.

Author Response
REVIEWER 2
Major Remarks:
- It is not clear whether the authors used recommended correction factor for the pulse wave
velocity (i.e. 0.8). This is serious methodological limitation of the study. The authors are
encouraged to address this issue.
Thank you for this observation. The Pulse Pen software we used automatically multiplies the carotid-femoral distance by a factor of 0.8, as suggested by the 2012 expert consensus document. This issue is now underlined in the method section. (Line 110-111)
- Other important parameters of arterial stiffness were not reported, such as central and
peripheral augmentation index (corrected for the heart rate of 75 bpm). This is serious
methodological limitation of the study. The authors are encouraged to address this issue.
We agree with the reviewer: augmentation index is deemed to be an important parameter, yet, since it was not significant in our findings, we decided to exclude it from the results. This has been added in the new version of the paper as a limitation of the study (lines 229-230).
- It is unclear why do the authors compare baseline characteristics between the male and
female patients when this was not the aim of the study, nor does it have any reasonable
background to be used in this study. It only introduces confusion to the readers, and
increases the possibility of type 1 error with multiple unfocused analyses. If the authors
want to assess the interaction of sex, it is possible by doing interaction analyses, or
sensitivity analyses, or adjustments for sex in the multivariable analysis. If the authors aimed
to present this data in the table 1, it doesn’t need to consume too much text in the Results
section which is the case.
Thank you for this observation: according to the suggestion, those results have been excluded from the main text.
- The authors are encouraged to explain non-negligible variation in the sample age (67.5, SD
14.3 years). And particularly having in mind that the sample was enrolled in the Geriatric
clinic. This substantial age variation (large proportion of patients was <65 years old) is
introducing uncertainty to the readers.
Thanks for the opportunity of clarifying this issue. Patients have been enrolled both from the Geriatric Ward (aged more than 70 years old) and from the Outpatient Clinic, which is affiliated to the Geriatric Department but eligible for patients of any age (patients referred due to Nutritional or Hypertension issues). This aspect is now added to the method section (line 74). Since the aim of our study was to depict the arterial stiffness indexes throughout aging, we decided to include a large cohort of subjects, choosing a broad age range.
- There is no point in assessing the correlation between CAVI and CAVI0. Or the correlation
between DBP and age. It is not the aim of this study, it is redundant, and it only introduces
confusion to the readers, as well as increases the possibility of type 1 error with multiple
unfocused analyses. Please focus your analyses based on the stated aims – correlation of AS
parameters with age and comorbidity burden, including the unadjusted, adjusted and
sensitivity analyses.
We agree with the reviewer and major revision was performed to follow this recommendation; the comparison between CAVI and CAVI 0 has been removed by this manuscript, focusing our attention on the correlation of arterial stiffness parameters with age and comorbidity burden. New regression models have been shown in Table 3 and in the results section. (line 157-165)
Minor Remarks:
- The authors are encouraged to use the appropriate terms for their results/findings that are
in accordance with the used statistical methods. Having in mind that the authors determined
linear correlation between the continuous quantitative variables, it is advised to use the
term „correlation“ instead of the term „association“
Revision was made to use the appropriate terms, thanks for this observation.
- Title of the manuscript needs to be revised. It is too ambiguous at the current form. I suggest
the following: The correlation of arterial stiffness parameters with age and comorbidity
burden amongst older adults.
We agree with the reviewer, the title has been changed accordingly.
- CAVI0 is not explained in the Abstract section.
CAVI 0 definition is now present in the abstract (line 31-32).
- Abstract section is too confusing
Abstract section has been revised to make it as clear as possible (line 30-41).
- Introduction section needs to be improved, with focus on the topic (arterial stiffness with
regards to age and comorbidity burden). The aims should be explained in a clear way. It
must be clearly stated that the aim of the manuscript is to determine the correlation between arterial stiffness parameters and age/comorbidity burden (defined as CCI). It is not
relevant to discuss differences between CAVI and CAVI0, etc. Overall, the Introduction
section needs to be substantially improved with the focus on the study goals and
background.
Revision was made to improve the whole Introduction Section (page 3). The difference between CAVI and CAVI 0 has been removed; the introduction is focused on our main purpose, which is to determine the correlation between arterial stiffness indexes, comorbidity burden (quantified as CCI) and cardiovascular risk factors in a group of adults and older adults.
- The authors are not consistently outlining the direction of the association, i.e. if it is positive
or negative association. Please address where necessary.
The associations have been better explained as required, to outline the direction of the relationships.
- The Results section would benefit from the subheadings.
Thanks for this recommendation, subheadings have been added.
- Figure 1 (A-C) needs to be modified by improving the presented data – it must be clearly
shown to the readers what is the row/column variable (CCI vs. AS parameters).
As suggested, figure 1 has been modified and caption has been improved.
- Lines 216-220: Please stated clearly your findings – the manuscript investigated the
correlation of comorbidity burden by CCI score, and not CV risk factors. Furthermore, the
manuscript was not designed to recommend superiority of CAVI0 over CAVI but
demonstrated a simple correlation between AS parameters and different factors such as age
and comorbidity burden (CCI score). Both parameters correlated well, and it was not part of
the study aims.
Thank you for this observation: the comparison of CAVI and CAVI 0 has been withdrawn by the manuscript. In our opinion, cardiovascular risk factors may significantly affect the comorbidity burden and their evaluation was actually a relevant part of our investigation. This issue has been better explained in the methods and results sections (line 77, line 145-154).
- Please focus your conclusions section – it needs to be clear and concise, without too much
text. The manuscript demonstrated significant correlation of PWV, CAVI and CAVI0 with age
and comorbidity burden. Further studies are needed to determine its relevance to the
clinical outcomes and prognosis of this population.
We agree with the reviewer and the Conclusion section has been modified (231-236).
- It is important to explain how the CCI score was calculated – using discharge ICD codes or
using the patient-reported data. It must be included in the Methods section.
We agree with the reviewer: our CCI is based on anamnestic patient-reported information, and this point is now described in the method section (Line 78-79).
- The following conclusion of the Abstract section is not supported by the study findings:
„Given the dependence of CAVI0 on DBP (lower in the elderly), CAVI may be more accurate
than CAVI0 to evaluate arterial stiffness among elderly individuals. “
As previously mentioned, the choice of CAVI instead of CAVI 0 is no longer discussed in the revised manuscript.
- The authors are encouraged to improve the use of abbreviations. Whenever the
abbreviation is 1 st mentioned, it needs to be spelled out, and later used only as abbreviation.
This is not the case in the current manuscript form. Please revise it appropriately and
provide evidence.
Abbreviations have been checked and modified as necessary.
- The age of the sample is redundantly repeated in the manuscript. It must be done only in the
Abstract section and the Results section – there is no need for repetition.
The age was removed by the method section (line 73).
- It is mandatory to improve the formatting of the manuscript. It contains a lot of lapses,
double spaces and grammatical errors which need to be corrected. Also, the language and
the flow of the manuscript needs substantial revision and improvement. Some of the
examples are outlined below:
o Bioumoral – line 14
o Comma is missing after CCI in line 17
o Fullstop is missing after the DBP interaction in line 23
o Double space – line 40; line 111
o CAVI 0 vs. CAVI0 must be consistently used (not interchangeably)
o Numbers such as Pearson’s r need to be rounded to 3 decimals consistently (there is
no place for interchangeable use of 2 vs. 3 decimals)
The text has been carefully checked out and those mistakes have been corrected, thank you for your corrections.

Round 2
Reviewer 2 Report
Please find attached the Review report. Best regards.
